# The non-specific phospholipase C of common bean *PvNPC4* modulates roots and nodule development

Ronal Pacheco[1], M.A. Juárez-Verdayes[2], A. I. Chávez-Martínez[1], Janet Palacios-Martínez[1], Alfonso Leija[3], Noreide Nava[1], Luis Cárdenas[1], Carmen Quinto[1]*

**1** Departamento de Biología Molecular de Plantas, Instituto de Biotecnología, Universidad Nacional Autónoma de México, Avenida Universidad, Colonia Chamilpa, Cuernavaca, Morelos, Mexico, **2** Departamento de Ciencias Básica, Universidad Autónoma Agraria Antonio Narro, Calzada Antonio Narro, Saltillo, Coahuila, Mexico, **3** Centro de Ciencias Genómicas, Universidad Nacional Autónoma de México, Av. Universidad, Mexico City, Mexico

* carmen.quinto@ibt.unam.mx

## Abstract

Plant phospholipase C (PLC) proteins are phospholipid-degrading enzymes classified into two subfamilies: phosphoinositide-specific PLCs (PI-PLCs) and non-specific PLCs (NPCs). PI-PLCs have been widely studied in various biological contexts, including responses to abiotic and biotic stresses and plant development; NPCs have been less thoroughly studied. No PLC subfamily has been characterized in relation to the symbiotic interaction between Fabaceae (legume) species and the nitrogen-fixing bacteria called rhizobia. However, lipids are reported to be crucial to this interaction, and PLCs may therefore contribute to regulating legume–rhizobia symbiosis. In this work, we functionally characterized *NPC4* from common bean (*Phaseolus vulgaris* L.) during rhizobial symbiosis, findings evidence that NPC4 plays an important role in bean root development. The knockdown of *PvNPC4* by RNA interference (RNAi) resulted in fewer and shorter primary roots and fewer lateral roots than were seen in control plants. Importantly, this phenotype seems to be related to altered auxin signaling. In the bean–rhizobia symbiosis, *PvNPC4* transcript abundance increased 3 days after inoculation with *Rhizobium tropici*. Moreover, the number of infection threads and nodules, as well as the transcript abundance of *PvEnod40*, a regulatory gene of early stages of symbiosis, decreased in *PvNPC4*-RNAi roots. Additionally, transcript abundance of genes involved in autoregulation of nodulation (AON) was altered by *PvNPC4* silencing. These results indicate that *PvNPC4* is a key regulator of root and nodule development, underscoring the participation of PLC in rhizobial symbiosis.

## Introduction

Rhizobial symbiosis is established between legume species (*Fabaceae*) and nitrogen-fixing soil bacteria called rhizobia. This interaction normally occurs in

**Data availability statement:** All relevant data are within the manuscript and its Supporting Information files

**Funding:** This research was partially funded by the National Council of Humanities, Science and Technology of Mexico (CONAHCyT) under Frontiers of Science (FC) CF-2023-I-297 to C. Quinto. CONAHCyT also funded a PhD fellowship to R. Pacheco (749422), a postdoctoral fellowship to J. Palacios Martínez (2342655), and a postdoctoral fellowship to A. I. Chávez Martínez." The Program to Support Research and Technological Innovation Projects/UNAM (PAPIIT) awarded grants N203021 and IN204024 to C. Quinto and IN204024 to R. Pacheco. The funders had no role in the study design, data collection and analysis, decision to publish, or preparation of the manuscript.

**Competing interests:** The authors have declared that no competing interests exist.

nitrogen-poor soils, when roots exude flavonoids into the rhizosphere that are specifically recognized by rhizobia. The specific recognition of flavonoids induces the synthesis and secretion by rhizobia of lipochitooligosaccharides known as nodulation factors (NFs), which are perceived by receptors at the plasma membrane of root hairs [1,2]. The perception of NF triggers the so-called common symbiotic pathway, which allows rhizobial infection and the initiation of nodule organogenesis [3]. The most common route of rhizobia infection occurs through the formation of a tubular structure known as the infection thread (IT) at the apex of root hairs. Bacteria migrate through the IT to the growing nodule and differentiate into bacteroids, giving rise to nitrogen-fixing organelles called symbiosomes. In the nodules, the nitrogen fixed (in the form of ammonia) by the bacteroids is exchanged for the carbon produced by the plants, providing a source of energy for the bacteria [4].

The establishment of rhizobial symbiosis involves several biological processes, such as: hormone-mediated signaling [5], reactive oxygen species signaling [6,7], autoregulation of nodulation [8–10], membrane turnover, and lipid-mediated signaling and metabolism [11,12]. The involvement of lipids in the establishment of this interaction has been reported to be crucial on the basis of pharmacological studies [13–15]. For example, phospholipids are thought to play a role in the NF-induced signal transduction pathway [15]. Moreover, lipid turnover in membranes has been demonstrated to be essential for nodulation [11]. In this scenario, phospholipases appear to be key regulators of symbiosis, as they hydrolyze phospholipids, the main components of membranes. Plant phospholipases are divided into three classes: A, C, and D. Specifically, phospholipases C (PLCs) cleave the glycerophosphate bond, releasing diacylglycerol and the phosphorylated head group [16]. Based on their substrate specificity, PLC members are classified into two groups or subfamilies: 1) phosphoinositide-specific PLCs (PI-PLCs) and 2) non-specific PLCs (NPCs) that non-specifically hydrolyze major membrane phospholipids such as phosphatidylcholine (PC) and phosphatidylethanolamine (PE) [17].

The structure of PI-PLC is characterized by the presence of two catalytic domains, PLC-X and PLC-Y; an EF-hand calcium-binding motif at their N terminus; and a C2 calcium-binding domain at their C terminus [18]. However, the EF-hand motif has not been reported in all PI-PLCs [19,20]. Unlike PI-PLCs, NPCs have a single catalytic domain, the phosphoesterase domain, and no EF-hand motif [19–21]. The functions of NPCs have been studied in abiotic stress [22], in plant immunity [23], and during plant development, specifically root development [24,25]. Their participation in rhizobial symbiosis remains unexplored to date. In fact, neither PI-PLC nor NPC from common bean (*Phaseolus vulgaris* L.) have been functionally characterized. In this work, we performed an *in silico* analysis of the PLCs in common bean (PvPLCs) and functionally characterized the *PvNPC4* gene during bean root development and in the symbiosis with *Rhizobium tropici*.

## Materials and methods

### *In silico* analysis of phospholipase C (PLC) family members

To identify members of the PLC family in common bean (*Phaseolus vulgaris* L.) and barrelclover (*Medicago truncatula* Gaertn.), a search was performed on the

corresponding genome sequences (v2.1-common bean and Mt4.0v1, respectively) available at the Phytozome 13 database [26] (https://phytozome.jgi.doe.gov/pz/portal.html) using the BLASTP algorithm. *Lotus japonicus* (Regel) K. Larsen, PLCs were identified in Lotus Base (https://lotus.au.dk/blast/#) MG20 v3.0. For BLASTP, the PLC protein sequences previously reported in Arabidopsis (*Arabidopsis thaliana* (L.), Heynh. were used as queries [27,28]. Additionally, the previously reported soybean (*Glycine max*) (L.) Merr. PLC protein family was searched and any missed proteins from the published list were added [21,22]. Conserved domains in putative PLC members were identified using SMART (http://smart.embl-heidelberg.de/) [29]. For phylogenetic analysis, the PLC protein sequences of Arabidopsis, soybean, and common bean were used. Sequences were aligned using the MUSCLE algorithm and manually edited using MEGA v. X [30] to remove misaligned sequences. The phylogenetic tree was reconstructed using the IQ-TREE algorithm version 1.6.12 [31] and the maximum-likelihood method based on JTT + G model with 10,000 bootstrap replicates.

Physicochemical parameters were predicted using ExPASy (https://web.expasy.org/cgi-bin/protparam/protparam) [32]. The subcellular localization of PvPLCs was predicted using WOLF-PSORT (https://wolfpsort.hgc.jp/), CELLO ver.2.5 (http://cello.life.nctu.edu.tw/) [33], and LocTree3 (https://rostlab.org/services/loctree3/) [34]. MEME ver. 5.4.1 (https://meme-suite.org/meme/tools/meme) [35] was used to identify conserved motifs from PvPLCs with the maximum number of motifs of 10, and the other parameters were set to default. The gene structure was analyzed by TB tools software [36].

The chromosomal position of *PvPLC*s was determined according to the Phytozome v13 database, and the chromosome distribution was visualized using TB tools software. The One-Step MCScanX-SuperFast program of TB tools software was used to perform pairwise duplication and synteny analysis. The *cis*-elements in the promoter regions (2000 bp upstream of the translation start codon) were analyzed using the PlantCARE online tool (https://bioinformatics.psb.ugent.be/webtools/plantcare/html/) and plotted using TB tools software. The nonsynonymous ($K_a$) and synonymous ($K_s$) substitution rates of each gene pair were calculated using TB tools software.

The expression profiles of *PvPLC* genes in various plant tissues at different developmental stages were retrieved from the *P. vulgaris* gene expression atlas database PvGEA (https://www.zhaolab.org/PvGEA/) [37]. Data were downloaded as reads per transcript kilobase per million reads mapped (RPKM). Heatmaps representing the expression profile of *PvPLC*s were plotted using TB tools software.

## Growth conditions of wild-type plants

Seeds of wild-type *P. vulgaris* L. cv. Negro Jamapa plants were germinated according to the protocol published at dx.doi.org/10.17504/protocols.io.261ge3bpjl47/v3 [38].

Two-day-old seedlings were planted in pots containing autoclaved vermiculite and watered every 3 days with sterile deionized water for 1 week. After 1 week, seedlings were grown in pots containing autoclaved vermiculite and inoculated with *Rhizobium tropici* CIAT 899 [39] at an $OD_{600}$ of 0.05 or not inoculated as control. Control plants were watered every 3 days with a solution of B & D medium [40] containing 2 mM $KNO_3$ and 0.015 mM $(NH4)_2SO_4$ to prevent rhizobial infection. Similarly, inoculated plants were watered every 3 days with nitrate-free B & D solution [40].

Roots samples were collected at 12, 24, and 72 h post inoculation (hpi) and at 14 days post inoculation (dpi). All tissues were stored at −75°C until analysis. The roots of 14 dpi plants were free of nodules. Additionally, leaves, stems, and roots were collected from seedlings irrigated with deionized water, at 7 days after transplanting into pots, and root hairs from 2-day-old seedlings. These tissues were used for total RNA extraction and quantification of *Pv-PI-PLC4* or *PvNPC4* transcript abundance by reverse-transcription quantitative PCR (RT-qPCR), as described below. The transcript abundance of *PvNPC4* in seedling tissues 7 days after transplanting to pots, was compared with the transcriptional profile of *PvNPC4* in different tissues and stages obtained from the transcriptional landscape https://www.zhaolab.org/PvGEA/ [37].

## RNA extraction and RT-qPCR assays

Total RNA extraction and complementary DNA synthesis were performed following the protocol published at dx.doi.org/10.17504/protocols.io.8epv5jq24l1b/v1. qPCR assays were conducted with a Maxima SYBR Green/ROX qPCR Master Mix (2X) kit (Thermo Scientific, Waltham, MA, USA) on a qPCR system (QuantStudio 5; Applied Biosystems, Waltham, MA, USA). The qPCR steps consisted of 95°C for 10 min, 30 cycles of 95°C for 15 s, and 60°C for 60 s. Relative transcript abundance were calculated with the $2^{-\Delta\Delta C_T}$ method [41] using *elongation factor 1α* (*EF1α*, Phvul.004G075100.1) and *β-tubulin* (Phvul.009G017300.1) as reference genes. The primer sequences used in all qPCR assays are given in S1 Table.

## Plasmid construction

For knockdown of *PvNPC4*, an RNAi silencing construct (*PvNPC4*-RNAi) was generated. To generate the *PvNPC4*-RNAi construct, a specific 131-bp fragment from the *PvNPC4* coding sequence was amplified using a common bean root cDNA as a template. This amplicon was inserted into the pENTR™/D-TOPO™ vector generating the donor vector. The donor vector was recombined with the binary vector pTDT-DC-RNAi [42] using GATEWAY™ technology to yield the *PvNPC4* silencing construct (S1 Fig). The correct orientation of the inserted fragment in the donor and binary vector was verified by PCR and Sanger sequencing using the appropriate primers (S1 Table). The pTdT-SAC vector carrying a truncated and unrelated sequence from Arabidopsis pre-*MIR159* was used as control [43].

## Generation of composite plants and selection of *Agrobacterium rhizogenes* clones of interest

Transgenic hairy roots were generated in wild-type plants of Negro Jamapa (composite plants) following the protocol described at dx.doi.org/10.17504/protocols.io.261ge3bpjl47/v3 [38]. Selection of *A. rhizogenes* clones for *PvNPC4*-RNAi knockdown was performed according to the criteria discussed previously [38]. To generate hairy roots, one *A. rhizogenes* clone transformed with the control vector was used, as well as several *A. rhizogenes* clones transformed with the *PvNPC4* silencing construct. The roots of common bean plants were inoculated with *R. tropici* CIAT899 or watered with B & D medium only and harvested at 10 dpi for RNA extraction, as described previously dx.doi.org/10.17504/protocols.io.8epv-5jq24l1b/v1. The silencing efficiency of *PvNPC4* by the RNAi construct was confirmed by RT-qPCR analysis; two *A. rhizogenes* clones transformed with the RNAi construct able to reduce *PvNPC4* transcript abundance in more than 60%, were selected (S2 Fig).

## Phenotypic analysis

The phenotypic analysis was performed on roots inoculated or not with *R. tropici* CIAT899 or *R. tropici* transformed with the β-glucuronidase gene (*R. tropici*-GUS). To examine the effect of *PvNPC4* silencing on root development, the length of primary roots was measured, and the number of primary and lateral roots were counted. To examine the effect of *PvNPC4* silencing on rhizobial symbiosis, the number of ITs was quantified in hairy roots at 10 dpi with *R. tropici*-GUS, as well as the number of nodules at 14 dpi with *R. tropici* CIAT899.

## Analysis of gene expression in transgenic roots

To search for the role of *PvNPC4* in regulating root development, we analyzed the transcript abundance of the auxin transporter gene *Pin-formed 1* (*PvPin1b*, Phvul.004G150600.1) [44] and two genes, *LATERAL ORGAN BOUNDARIES-DOMAIN 16 (LBD16)/ASYMMETRIC LEAVES2-LIKE 18 (ASL18)* (*PvASL18a*, Phvul.001G159300.1 and *PvASL18b*, Phvul.001G159300.1) by RT-qPCR. *PvASL18a* and *PvASL18b* were identified by BLAST-P in the Phytozome 13 [26] database using the LOB Domain IPR004883 sequence from AT2G42430.1. RT-qPCR analysis was performed using mRNA from hairy roots of uninoculated 10-day-old seedlings, carrying the *PvNPC4*-RNAi construct and the control plasmid. Hairy roots were collected from inoculated seedlings at 10 dpi for the quantification of genes related to rhizobial

symbiosis: *Nodule Inception* (*PvNIN*, Phvul.009G115800.1), *Early nodulin 40* (*Enod40*, Phvul.002G064166.1[45]), *To Much Love* (*PvTML1a*, Phvul.001G094400.1), and *Rhizobia-Induced CLE1* and 2 (*PvRIC1*, Phvul.005G096901.1; *PvRIC2*, Phvul.011G135900.1 [10]). *PvTML1a* was identified in the common bean genome available in Phytozome 13 [26] using the Kelch repeat domain sequence of F-box IPR052439 of AT3G27150.1 and AT5G40680.1. Primers for qPCR assays are listed in S1 Table.

## Statistical analysis

For statistical analysis, a bootstrap test with 9999 samplings with replacement and a Monte Carlo simulation test with 9999 samplings without replacement were used. These analyses require fewer assumptions than traditional methods and are more accurate than classical statistical analyses [46,47]. These methods make it possible to compare the value of a given statistic with a reference distribution generated from the data itself. Importantly, the Monte Carlo simulation is useful to estimate the *p*-value when the sample size is small [48], which was commonly the case in this study. Both tests were performed in RStudio 4.1.2 and 4.3.1 (http://www.rstudio.com/) using the tidyverse package [49]. RAWGraphs (https://app.rawgraphs.io/) was used to identify outliers in all datasets. Spearman correlation between *PvNPC4* transcript abundance in common bean tissues and its expression profile available in PvGEA obtained by RNA-seq [50] was performed in RStudio 4.3.

## Results

### Identification of PLCs in legume species

To evaluate the putative role of PLCs in the symbiotic relationship between legumes and rhizobia, we asked whether *PLC* genes constitute a gene family in the genomes of several model legumes, such as barrelclover, *L. japonicus*, and common bean. To this end, we queried the corresponding legume genome databases using Arabidopsis protein sequences as queries. After discarding redundant sequences and confirming conserved domains using SMART, we identified 15 non-redundant PLCs in barrelclover and 12 PLCs each in *L. japonicus* (S2 Table) and common bean (S3 Table). The same analysis returned 13 PI-PLC members, instead of the published number of 12, and 9 NPC members rather than the published number of 7 in the soybean genome [21,22]; we provide the updated list of GmPLCs in S2 Table.

In common bean, the amino acid sequences of seven PvPLCs showed the characteristic domains of the PI-PLC subfamily, while the remaining five PvPLCs displayed the phosphoesterase domain associated with members of the NPC subfamily (S3 Fig). Moreover, all putative PvPI-PLCs above presented the catalytic PLC_X and PLC_Y domains, as well as the $Ca^{2+}$ and phospholipid-binding domains; we identified the EF-hand motif in three of the seven PvPI-PLCs. Furthermore, only PvPI-PLC5 had a predicted signal peptide (S3 Fig). For the predicted PvNPCs, all had the characteristic phosphoesterase domain, and only PvNPC4 lacked a predicted signal peptide (S3 Fig). We named all PLCs based on their subfamily classification and chromosomal location (S3 Table).

### PLCs of legume species are phylogenetically close

To analyze the function of PvPLCs in the symbiotic interaction between legumes and rhizobia, we reconstructed a phylogenetic tree using the amino acid sequences of all model legume species used for the identification of PLC members. We also included non-legumes as reference: Arabidopsis, a non-legume dicotyledonous species, and rice (*Oryza sativa* L.), a monocotyledonous species. Phylogenetic reconstruction separated all these PLCs into two well-defined clades, corresponding to PI-PLCs and NPCs (Fig 1). Importantly, NPCs clustered closer to the outlier sequence used here, a *Mycobacterium tuberculosis* PLC sequence, than to the PI-PLC clade. Within the PI-PLC and NPC clades, PLCs from non-legume species (Arabidopsis and rice) were the most phylogenetically distant from those from legume species. Importantly, PLCs from legume species were more closely related to each other than to non-legumes, with the smallest

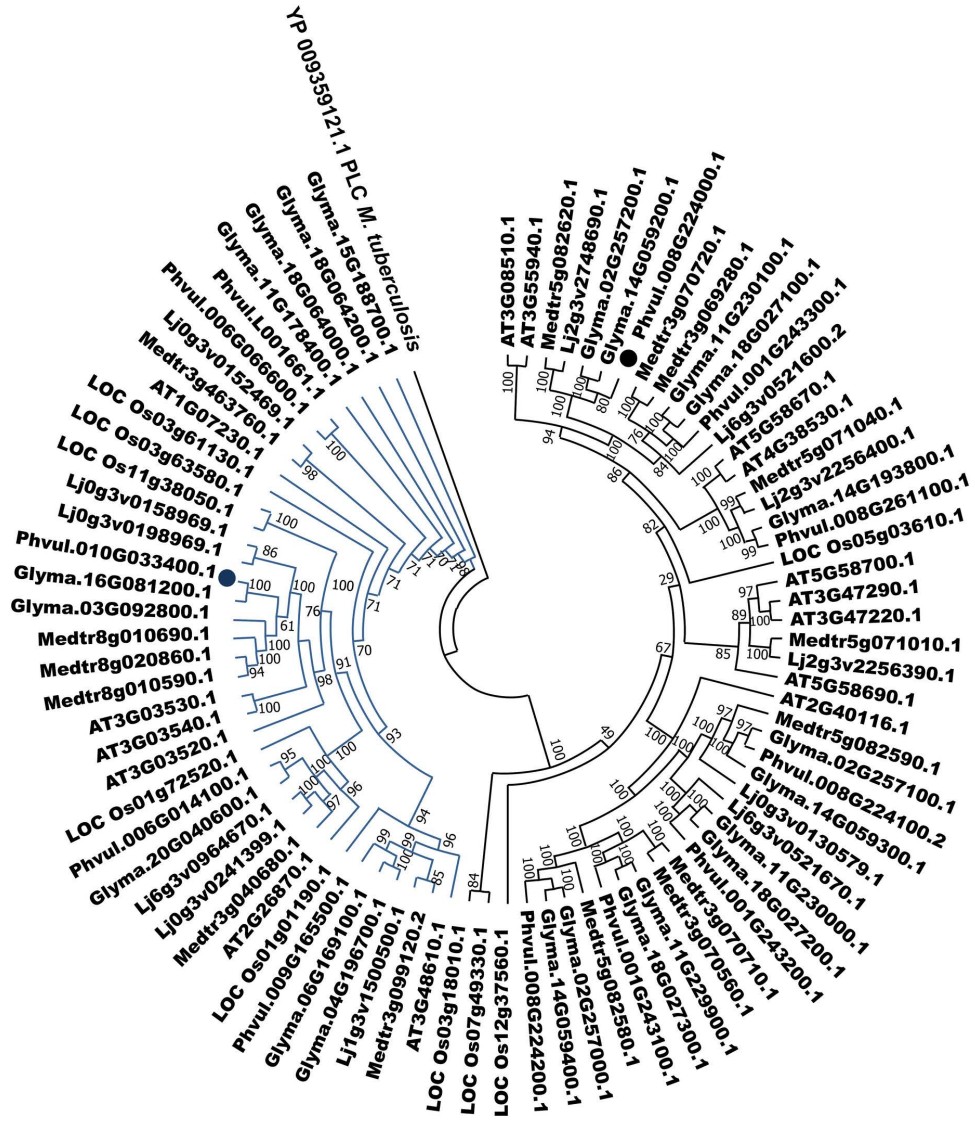

**Fig 1. PLCs of leguminous and non-leguminous species are grouped into two clades.** The plant species used in this analysis are *M. truncatula* (Medtr), *L. japonicus* (Lj), *G. max* (Glyma), *P. vulgaris* (Phvul), *A. thaliana* (AT), and *O. sativa* (LOC Os). Blue indicates members from the PI-PLC clade; green indicates members from the NPC clade. The amino acid sequence of *M. tuberculosis* PLC (YP_009359121.1) was used as an outlier. The phylogenetic tree was reconstructed using the IQ-TREE algorithm based on the maximum-likelihood method and the JTT + G model; 10,000 bootstraps were performed.

phylogenetic distance being between common bean and soybean (Fig 1 and S4 Fig). Together, these results suggest an ancient evolutionary divergence between legume and non-legume PLCs.

To better typify PvPLCs, we analyzed some of their physicochemical properties (S4 Table). PvPI-PLC members were longer than PvNPCs, ranging from 550 to 636 amino acids compared to 487–525 amino acids for PvNPCs and therefore had higher molecular weights. Furthermore, PvPI-PLCs showed a heterogeneous isoelectric point (pI): four had a slightly acidic pI, one had a neutral pI, and two had a slightly alkaline pI. In contrast, four of the five PvNPCs showed a slightly acidic pI; only PvNPC1 had a marginally alkaline pI. The grand average of hydropathy (GRAVY) values for all PvPLCs were predicted to be less than 0, indicating their hydrophilic behavior.

The *PvPLC* subfamily members presented different exon-intron structures; their encoded proteins also displayed distinct protein motifs. We detected nine exons in *PvPI-PLC* genes and three or four exons in *PvNPC* genes (S5 A, B Fig). We identified no 3′ untranslated region (UTR) in *PvPI-PLC7* and neither 5′ UTR nor 3′ UTR in *PvNPC4* and *PvNPC5* (S5A, B Fig). In the PvPLC protein sequences, we identified 10 protein motifs; PvPI-PLCs contain motifs 1–8, while PvNPC members contain motifs 4, 5, 7, 9, and 10 (S5C Fig).

### *PvPLC* genes have undergone duplication

We assigned all *PvPLC* genes to chromosomes in the common bean genome, as described in Materials and methods. Eleven of the *PvPLC* genes were located on 5 of the 11 chromosomes of common bean as follows: three genes on chromosome 1, two genes on chromosome 6, four genes on chromosome 8, one gene on chromosome 9, and one gene on chromosome 10. The remaining gene currently maps to a scaffold (S6 Fig). The three genes on chromosome 1 and the four genes on chromosome 8 correspond to *PI-PLC* genes. The three genes on chromosome 1, as well as three of the four genes on chromosome 8, appear in tandem along the chromosome (S6 Fig). In contrast, *NPC* genes are widely distributed along the three chromosomes to which they map, with no evidence of tandem arrangement (S6 Fig).

To explore the evolutionary relationships between the *PvPLC* genes in common bean, we performed a synteny analysis using the program MCScanX. The analysis suggested two segmental duplication events corresponding to the *PvPLC1*–*PvPLC4* and *PvPLC1*–*PvPLC7* gene pairs and one tandem duplication event for the *PvPLC4*–*PvPLC7* gene pair (Fig 2A). We detected no collinearity among the *PvNPC* genes. When we calculated the ratios of nonsynonymous to synonymous changes ($K_a/K_s$) between *PvPLC* pairs, we obtained values below 1 (S3 File), suggesting that they underwent purification and selection after duplication. A synteny analysis between common bean and soybean identified 16 orthologous pairs, emphasizing the close evolutionary relationship between these two legume species (Fig 2B).

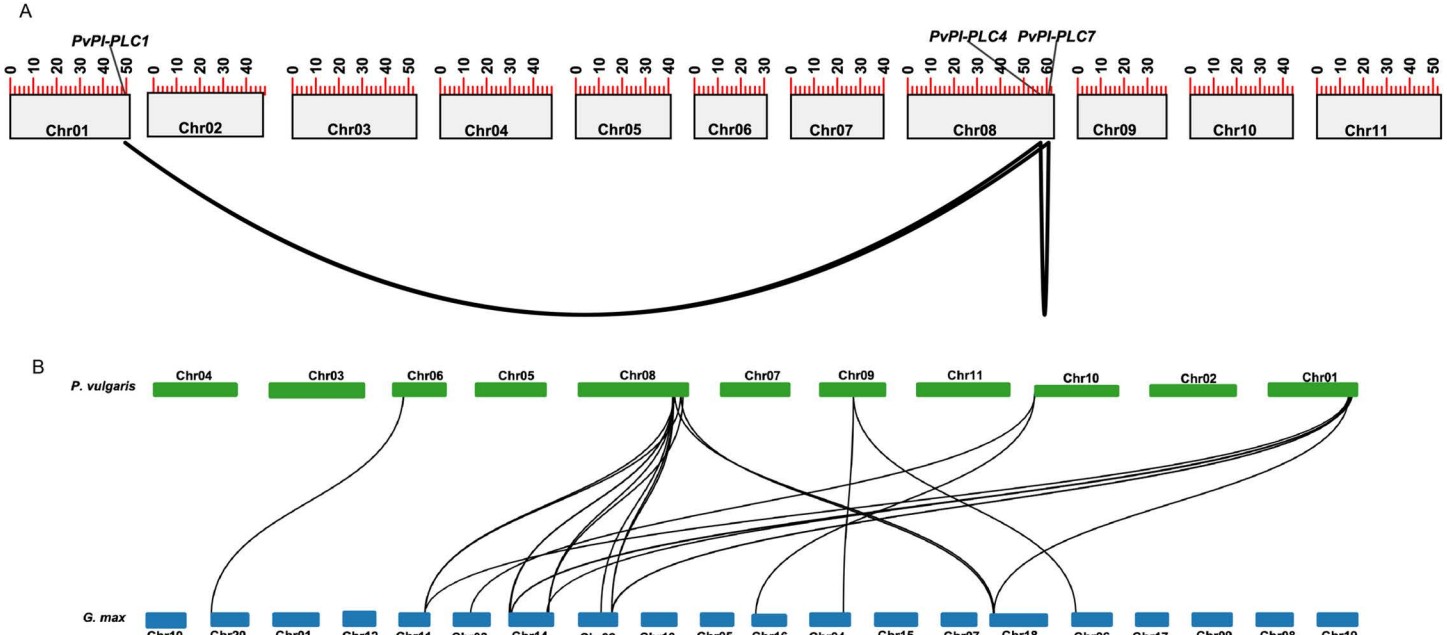

**Fig 2. *PvPLC* family contains gene duplication and has collinearity with *GmPLC* genes. (A)** *PvPLC* gene duplication analysis. **(B)** Collinearity analysis of *PLC* genes in common bean and soybean. Black lines show collinearity between *PvPLC* genes; red lines show collinearity between *PvPLC* and *GmPLC* genes.

## Identification of *cis*-regulatory elements in the promoter region of *PvPLC* genes

To better understand the regulation of *PvPLC* gene expression, we looked for *cis*-elements located in their promoter regions (up to 2000 bp upstream of the translation start site) using the online tool PlantCARE. We identified 57 *cis*-acting elements clustered into three groups: light responsive, phytohormone responsive, and stress responsive (S3 File). The *cis*-acting elements in the *PvNPC4* promoter were classified as follows: seven related to light and stress sensitivity, three to growth and developmental sensitivity, and eight to sensitivity to phytohormones (Table 1). Among the elements related to light, box-4 and G-box were the most represented; the main stress-related elements represented were anoxic responsive (ARE) and W-box. Among growth and development *cis*-elements, the CAT-box was present in the greatest number of copies, with TCA-element being the most frequent *cis*-element related to phytohormones (Table 1).

## Expression of *PvPI-PLC4* and *PvNPC4* genes varies depending on plant tissue and development and in response to rhizobia inoculation

We analyzed the expression patterns of genes belonging to the PvPLC family in different common bean tissues. Since the *PvPLC* family comprises several genes, we selected two genes, *PvPI-PLC4* and *PvNPC4*, for characterization in bean–rhizobia symbiosis. This selection was based on their expression profiles reported in the common bean gene expression atlas (PvGEA) as well as their phylogenetic relationship with Arabidopsis *PLC* genes. *PvPI-PLC4* showed the highest transcript abundance (in RPKM) in roots and nodules (S7 Fig). *PvNPC4* is phylogenetically closer to Arabidopsis *NPC4* (At3g03530) (S4 Fig) which is reported to be involved in the degradation of plasma membrane phospholipids [28]. Considering that phospholipid metabolism is crucial for nodule organogenesis [11,12], it was of interest to select this gene to study its role in rhizobial symbiosis. To examine the expression profile of these two common bean genes during symbiosis, we quantified their transcript abundance at different stages of symbiosis with *R. tropici* using RT-qPCR analysis.

Unexpectedly, we barely detected *PvPI-PLC4* transcript abundance in roots and nodules of common bean under our conditions. We observed no significant differences between inoculated and non-inoculated roots at 12, 24, or 72 h post inoculation (hpi), with *p*-values of 0.25, 0.86, and 0.7, respectively (S8A Fig). Likewise, *PvPI-PLC4* transcript was not significantly different at 14 dpi between inoculated roots and nodules compared to non-inoculated control roots, with *p*-values of 0.6 and 0.9, respectively. Moreover, no significant difference was observed in inoculated roots compared to nodules, with a *p*-value of 0.6 (S8B Fig).

Relative *PvNPC4* transcript abundance during rhizobial symbiosis did not change significantly in inoculated roots, at 12 or 24 hpi compared to non-inoculated roots (*p* = 0.28 and 0.354, respectively). However, at 72 hpi, relative *PvNPC4* transcript abundance significantly increased in inoculated roots compared to non-inoculated roots (*p* = 0.02) (Fig 3A). In contrast, relative *PvNPC4* transcript abundance significantly decreased in inoculated roots (*p* = 0.024) and nodules (*p* = 0.029)

**Table 1. Number and putative *cis*-elements identified in the *PvNPC4* promoter region (2000 bp upstream of the translation start codon).**

| Light sensitivity | AE-box | ATCT-motif | Box-4 | chs-CMA1a | chs-CMA2a | GATA-motif | G-box | |
|---|---|---|---|---|---|---|---|---|
| | 1 | 1 | 3 | 1 | 1 | 1 | 3 | |
| Stress sensitivity | ARE | as-1 | MYB | MYB-like sequence | MYC | STRE | W-box | |
| | 4 | 1 | 2 | 1 | 2 | 1 | 3 | |
| Growth and development | CAT-box | O2-site | TCA | | | | | |
| | 2 | 1 | 1 | | | | | |
| Phytohormone sensitivity | AAGAA-motif | ABRE | ABRE3a | ABRE4 | CGTCA-motif | ERE | TCA-element | TGACG-motif |
| | 1 | 1 | 1 | 1 | 1 | 2 | 3 | 1 |

AE-box, part of a module for light response; chs-CMA1a, part of light-responsive element; ARE, anoxic responsive; as-1, drought responsive; STRE, stress responsive; ABRE, abscisic acid responsive; ERE, ethylene responsive.

compared to non-inoculated roots at 14 dpi (Fig 3B). To further investigate the changes in *PvNPC4* transcript abundance upon rhizobium inoculation, we analyzed *PvNPC4* transcript abundance in inoculated and non-inoculated roots over time (12, 24, and 72 hpi, and 14 days post inoculation [dpi]).

In non-inoculated roots, *PvNPC4* transcripts showed two points of high abundance, at 12 h and 14 days, while decreasing between 24 and 72 h (S9A Fig). These two high points of transcript abundance were significantly different compared to the transcript abundance at 72 h ($p < 0.001$). In inoculated roots, we observed three high points of *PvNPC4* transcript abundance, at 24 hpi and 72 h, and decreased considerably in roots and nodules at 14 dpi (S9B Fig). Statistical analysis showed differences in inoculated roots of 12 dpi compared to roots of 14 dpi ($p = 0.04$) and nodules of 14 dpi ($p = 0.03$). Moreover, significant differences were observed in roots of 24 hpi compared to roots of 14 dpi ($p = 0.049$) and nodules of 14 dpi ($p < 0.001$).

Additionally, we quantified *PvNPC4* transcript abundance in leaves, stems, and roots of seedlings of 7 days after transplanting into pots irrigated with deionized water, and in root hairs of 2-day-old seedlings. We observed the lowest expression of *PvNPC4* in leaves and the highest in roots compared to the transcriptional landscape obtained from PvGEA (S10A Fig). Importantly, a Spearman correlation indicated that the transcript abundance obtained by qPCR and the PvGEA transcriptional profile obtained by RNA-seq are not significantly different (S10C Fig). Additionally, the transcriptional landscape of *AtNPC4*, the ortholog of *PvNPC4*, obtained from *Arabidopsis thaliana* Single Cell Atlas Data also showed the highest expression in roots (S10B Fig). Together, these findings suggest that *PvNPC4* may play an important role in roots by modulating its expression in response to rhizobial infection.

## Knockdown of *PvNPC4* negatively affects root development and nodule formation

To explore the involvement of *PvNPC4* during rhizobial symbiosis, we examined the nodulation phenotype in hairy roots carrying a *PvNPC4*-RNAi construct for *PvNPC4* knockdown, using hairy roots carrying the empty vector as a control. To this end, we used two independent *A. rhizogenes* clones carrying the *PvNPC4*-RNAi vector, namely, RNAi-C1 and RNAi-C8,

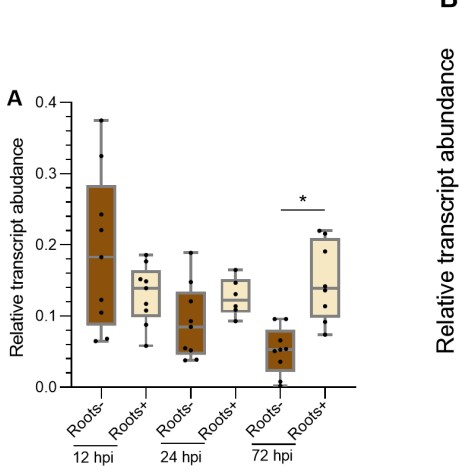
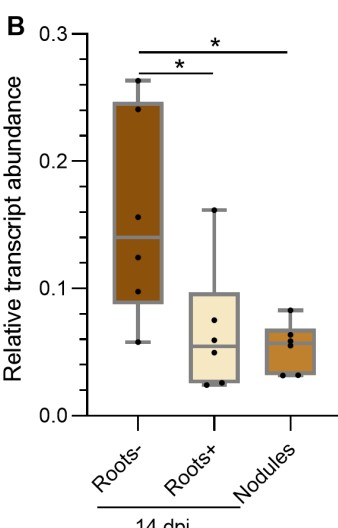

**Fig 3. *PvNPC4* transcript abundance changes differentially during rhizobial symbiosis. (A)** Relative abundance of *PvNPC4* transcript at early stages after inoculation with *R. tropici*. **(B)** Relative abundance *PvNPC4* of transcript at 14 dpi with *R. tropici*. Roots-, not inoculated roots; Roots +, inoculated roots without nodules. The lower and upper edges of boxes delimit the first to third quartiles, respectively, the central horizontal line represents the median, and the whiskers indicate the maximum and minimum values in the data set. Statistical significance was assessed with a Monte Carlo simulation test with 9999 resamples without replacement (* $p \leq 0.05$). Black dots in the box plots indicate independent samples from three biological replicates.

whose silencing efficiency was previously validated at 10 dpi with *R. tropici* (S2 Fig). Surprisingly, *PvNPC4*-RNAi seedlings developed much shorter and fewer roots compared to the control vector (Fig 4D). Indeed, the length of primary *PvNPC4*-RNAi roots was about half that of roots carrying the control vector, a difference that was significant ($p<0.001$) (Fig 4A). Similarly, *PvNPC4*-RNAi roots had fewer primary and lateral roots compared to the control (Fig 4B, C).

Auxin is a key regulator of root development [51], which prompted us to explore the relative transcript abundance of genes involved in the auxin-mediated signaling in 10-day-old transgenic hairy roots irrigated with B & D medium. Among the genes related to auxin signaling, *LBD16/ASL18*, has been shown to participate in root development [52]. Through *in silico* analysis, two *LBD16/ASL18* orthologs were identified in common bean, *PvASL18a* and *PvASL18b*, whose expression profiles in the PvGEA and in an Open Big Data metatranscriptome showed a high expression level in roots and nodules of common bean (S5 File). Regarding auxin transport, the common bean genome encodes 16 genes involved in this

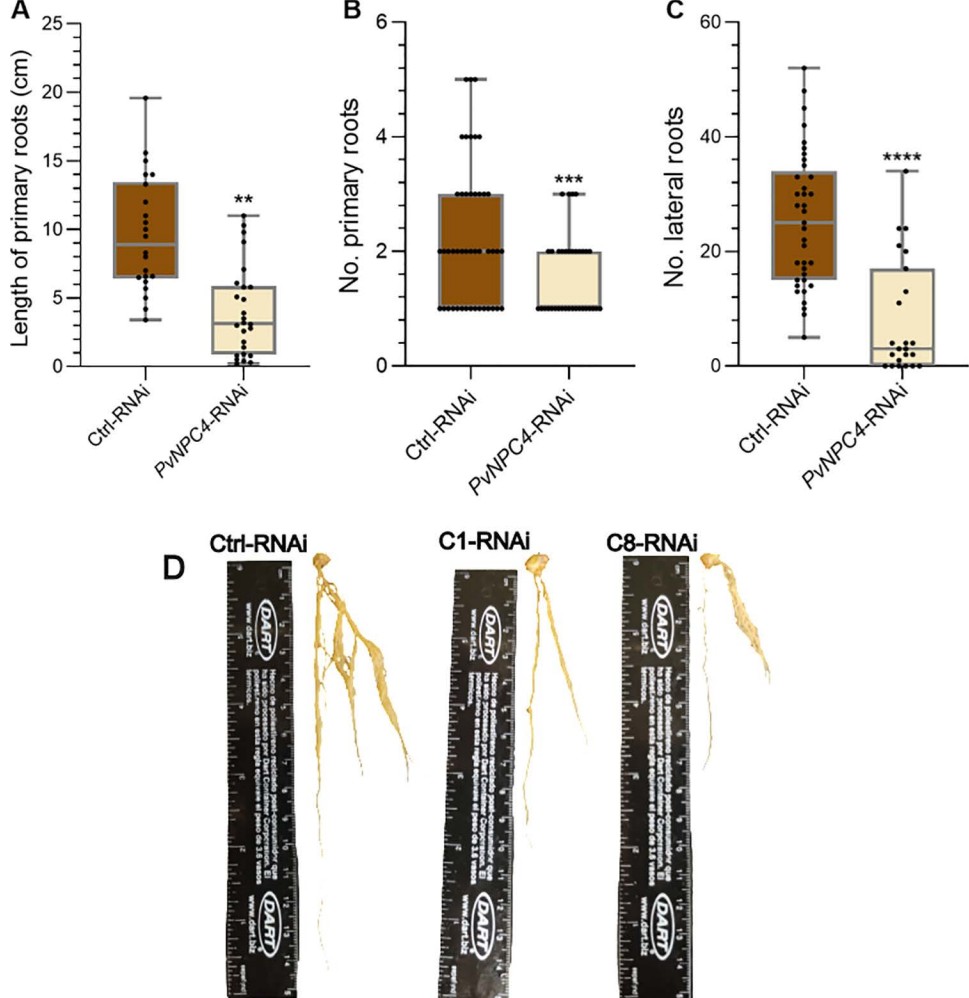

**Fig 4. *PvNPC4*-RNAi reduced root development of composite plants. (A)** Length (cm) of primary hairy roots, **(B)** number of primary hairy roots, and **(C)** number of lateral hairy roots, **(D)** transgenic roots carrying the RNAi control vector or the *PvNPC4* silencing construct. Ctrl-RNAi indicates roots carrying the control vector. *PvNPC4*-RNAi indicates transgenic roots carrying the *PvNPC4* silencing construct (C1-RNAi and C8-RNAi lines). The lower and upper edges of boxes delimit the first to third quartiles, respectively, the central horizontal line represents the median, and the whiskers indicate the maximum and minimum values in the data set. For statistical analysis, a bootstrap test was performed with 9999 samples with replacement (** $p \leq 0.01$, *** $p \leq 0.001$, **** $p \leq 0.0001$). Black dots in the box plots indicate individual samples from four biological replicates.

process [44], showing that the auxin transporter *PvPin1b*, is the most abundant transcript in roots and nodules according to the profile in PvGEA (S5 File).

*PvASL18a* and *PvASL18b* transcript abundance was slightly and significantly (*p* < 0.001) reduced, respectively, in *PvNPC4*-RNAi roots compared to control (Fig 5A). On the other hand, the abundance of *PvPin1b* transcripts decreased significantly (*p* = 0.02) by approximately 50% in *PvNPC4*-RNAi hairy roots compared to control roots (Fig 5B). These findings strongly suggest that *PvNPC4* in common bean is involved in root development in an auxin-dependent manner. This finding deserves further investigation; however, it is outside our focus regarding the role of the *PvNPC4* gene in the mutualistic interaction between common bean roots and rhizobia.

*PvNPC4* transcript abundance differentially changed in rhizobium-inoculated roots and in nodules at the early and mature stages of symbiosis (Fig 3). This observation prompted us to analyze the participation of PvNPC4 in nodule development. To this end, we quantified the number of ITs and the number of nodules at 10 and 14 dpi with *R. tropici*. Importantly, the number of ITs in *PvNPC4*-RNAi hairy roots at 10 dpi was significantly lower than that in the control (Fig 6A) (*p* < 0.001). In agreement with this result, the number of nodules at 14 dpi also decreased following *PvNPC4* silencing (*p* = 0.0067) (Fig 6B).

To better understand the regulatory function of PvNPC4 in rhizobial symbiosis, we quantified the transcript abundance of the two early nodulin genes *PvNIN* and *PvEnod40* in transgenic roots inoculated with *R. tropici* (10 dpi), as markers of symbiosis progression. We did not detect statistical differences in the transcript abundance of *PvNIN* in *PvNPC4*-RNAi roots compared to control roots (Fig 7A). However, *PvEnod40* transcript abundance was significantly decreased (*p* < 0.001) in *PvNPC4*-RNAi roots at 10 dpi (Fig 7B). Overall, these findings suggest that PvNPC4 participates in rhizobial symbiosis in common bean by modulating the expression of some early nodulins.

Previous studies by our group have shown the important role of NADPH oxidases, known in plants as RESPIRATORY BURST OXIDASE HOMOLOGS (RBOHs), in rhizobial symbiosis [6,7]. Moreover, PLCs and RBOHs have been shown to participate in plant response mechanisms in biotic interactions [23,53]. Therefore, we analyzed the transcript abundance of *PvRbohA*, a gene encoding a NADPH oxidase, in inoculated and non-inoculated *PvNPC4*-RNAi hairy roots at 10 dpi. No significant differences were observed between *PvNPC4*-RNAi roots and RNAi control roots in inoculated or non-inoculated roots (Fig 7C). In both inoculated and non-inoculated roots, *PvRbohA* transcript abundance was slightly

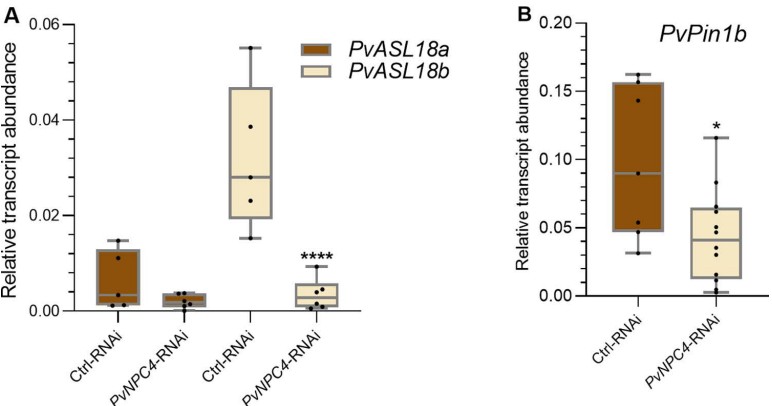

**Fig 5. Transcript abundance of the *PvASL18a, PvASL1b* and *PvPin1b* genes in transgenic roots of common bean silenced in the *PvNPC4* gene. (A)** Relative transcript abundance of *PvASL18a* and *PvASL18b*, **(B)** relative transcript abundance of *PvPin1b*. Ctrl-RNAi indicates transgenic roots carrying the control vector (pTdT-SAC). *PvNPC4*-RNAi indicates transgenic roots carrying the *PvNPC4* silencing construct (C1-RNAi and C8-RNAi lines). The lower and upper edges of the boxes delimit the first to third quartiles, respectively, the central horizontal line represents the median, and the maximum and minimum values in the data set. For statistical analysis, a Monte Carlo simulation test was used with 9999 samples without replacement (* *p* ≤ 0.05, **** *p* ≤ 0.0001). Black dots in the box plots indicate individual samples from three biological replicates.

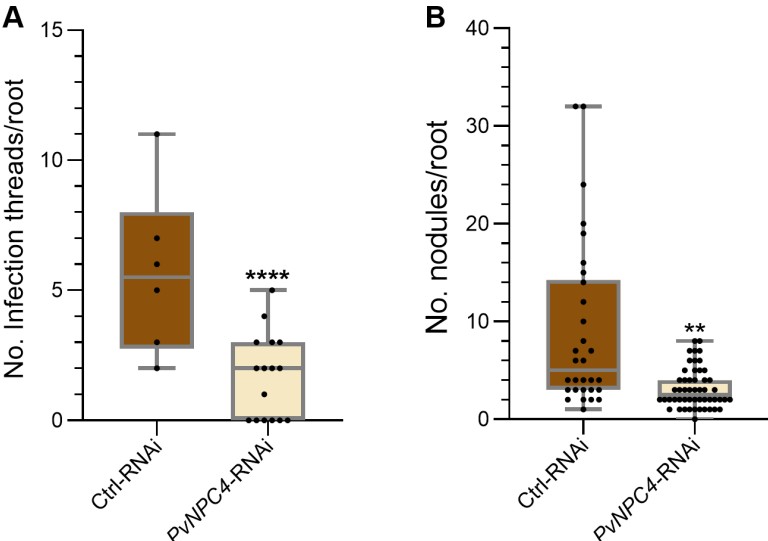

**Fig 6. *PvNPC4* silencing may regulate the numbers of infection threads and nodules in common bean hairy roots. (A)** Number of infection threads (ITs) per root at 10 dpi. **(B)** Number of nodules per root at 14 dpi. Ctrl-RNAi indicates transgenic roots carrying the control vector pTdT-SAC. *PvNPC4*-RNAi indicates transgenic roots carrying the *PvNPC4* silencing construct (C1-RNAi and C8-RNAi lines). The lower and upper edges of the boxes delimit the first to third quartiles, respectively, the central horizontal line represents the median, and the whiskers indicate the maximum and minimum values in the data set. For statistical analysis of IT numbers, a Monte Carlo simulation test was used with 9999 samples without replacement (**** $p \le 0.0001$; two biological replicates). For the number of nodules, a bootstrap test was performed with 9999 samples with replacement (* $p \le 0.01$; three biological replicates). Black dots in the box plots indicate individual samples.

increased in *PvNPC4*-RNAi roots compared to RNAi control roots. Importantly, *PvRbohA* transcript abundance was very similar between inoculated and non-inoculated *PvNPC4*-RNAi roots (Fig 7C), indicating that *PvNPC4* silencing can affect *PvRbohA* expression regardless of rhizobial infection.

To examine the possible role of *PvNPC4* in the autoregulation of nodulation (AON) pathway [54], transcript abundance of AON-related genes, such as *RIC* and *TML*, was quantified in control and *PvNPC4*-RNAi roots, non-inoculated or inoculated with *R. tropici* (10 dpi). It was previously reported that *P. vulgaris RIC1* and *RIC2* were induced by *R. phaseoli* in common white bean [10]. Along these same lines, we quantified the abundance of *PvRIC1* and *PvRIC2* transcripts in *PvNPC4*-RNAi and RNAi control roots, non-inoculated or inoculated with *R. tropici* (10 dpi). In non-inoculated roots, no statistical difference in *PvRIC1* transcript abundance was observed between control and *PvNPC4*-RNAi roots (Fig 8A). However, in inoculated roots, the transcript abundance of this gene was significantly induced ($p < 0.001$) in *PvNPC4*-RNAi roots compared to RNAi control roots (Fig 8A). Additionally, *PvRIC1* transcript abundance in inoculated *PvNPC4*-RNAi roots was higher ($p < 0.001$) than that in non-inoculated *PvNPC4*-RNAi roots (Fig. 8A) indicating that *PvNPC4* and *PvRIC1* are functionally interconnected during rhizobial symbiosis.

Regarding transcript abundance of *PvRIC2*, no statistical differences were found between *PvNPC4*-RNAi roots and RNAi control roots either in non-inoculated or inoculated roots (Fig 8B). Interestingly, *PvRIC2* transcript abundance in inoculated *PvNPC4*-RNAi roots was significantly reduced compared to that in non-inoculated *PvNPC4*-RNAi roots (Fig 8B), likely as a compensatory mechanism for *RIC* expression in this specific stage of common bean-*R. tropici* symbiosis. Transcription profiles of common bean *TML* genes were first examined using data from PvGEA and raw data available in Open Big Data. Through this analysis, *PvTMLa* was identified as having the highest transcript abundance in roots and nodules (S4 File). Using qPCR analysis, we detected a significant reduction in *PvTMLa* transcript abundance ($p < 0.001$) in *PvNPC4*-RNAi roots compared to control roots (Fig 8C). In contrast, in inoculated roots no differences were observed between *PvNPC4*-RNAi roots and control roots (Fig 8A).

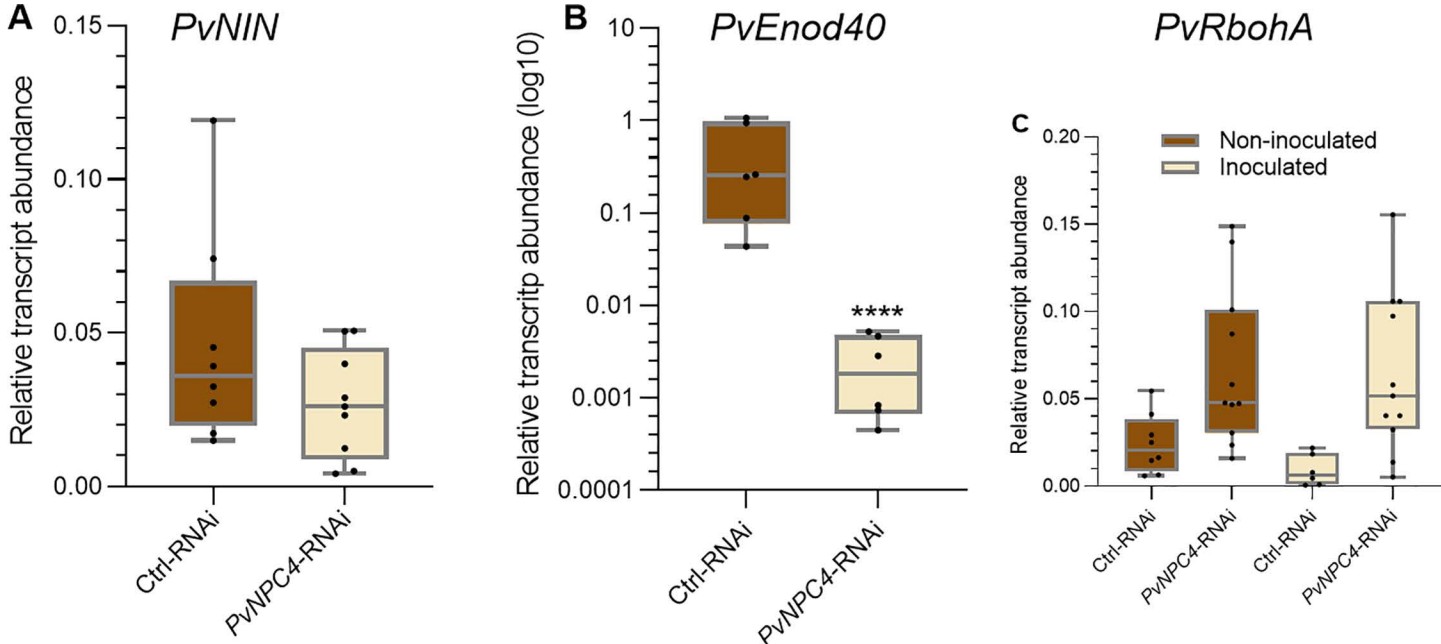

**Fig 7. *PvNPC4* silencing significantly reduces *PvEnod40* transcript abundance.** Relative transcript abundance of *PvNIN* **(A)**, *PvEnod40* **(B)**, and *PvRbohA* (C) at 10 dpi. Ctrl-RNAi indicates transgenic roots carrying the control vector pTdT-SAC. *PvNPC4*-RNAi indicates transgenic roots carrying the *PvNPC4* silencing construct (C1-RNAi and C8-RNAi lines). The lower and upper edges of boxes delimit the first to third quartiles, respectively, the central horizontal line represents the median, and the whiskers indicate the maximum and minimum values in the data set. For statistical analysis, a Monte Carlo simulation test was used with 9999 samples without replacement (**** $p \leq 0.0001$. Black dots in the box plots indicate individual samples from three biological replicates.

The *ASL18/LBD16* genes have been shown to be involved in root and nodule development [52,55], therefore, we examined the effect of *R. tropici* on the transcript abundance of these genes. For this, transcript abundance of *PvASL18a* and *PvASL18b* was quantified in *PvNPC4*-RNAi and control roots at 10 dpi and compared to non-inoculated roots. As observed previously (Fig 5A), in non-inoculated roots, *PvASL18a* transcript abundance did not change in *PvNPC4*-RNAi roots compared to RNAi control roots. In this analysis, no significant differences were found in inoculated *PvNPC4*-RNAi roots compared to inoculated RNAi control roots (10 dpi) (Fig 9A). Moreover, when comparing the abundance of the *PvASL18a* transcript in non-inoculated *versus* inoculated control roots, no significant difference was also observed. Likewise, *PvASL18a* transcript abundance did not change between non-inoculated and inoculated *PvNPC4*-RNAi roots (Fig 9A). On the other hand, *PvASL18b* transcript abundance did not change significantly in *PvNPC4*-RNAi inoculated roots compared to RNAi control roots (Fig 8B), as it did in non-inoculated roots (Fig 5B). Interestingly, *PvASL18b* transcript abundance in *PvNPC4*-RNAi inoculated roots significantly increased compared to *PvNPC4*-RNAi non-inoculated roots (Fig 9B).

## Discussion

The participation of non-specific phospholipases (NPCs) in plant development has not been thoroughly analyzed. Previous studies have mainly focused on Arabidopsis, where the role of NPC in root development was examined. Studies performed in other plant models as rice [19] and wheat (*Triticum aestivum* L,) [20] have limited their analysis to the expression of NPC during certain stages of development and abiotic stresses. Here, we showed that the common bean gene *PvNPC4* plays a key role in root and nodule development.

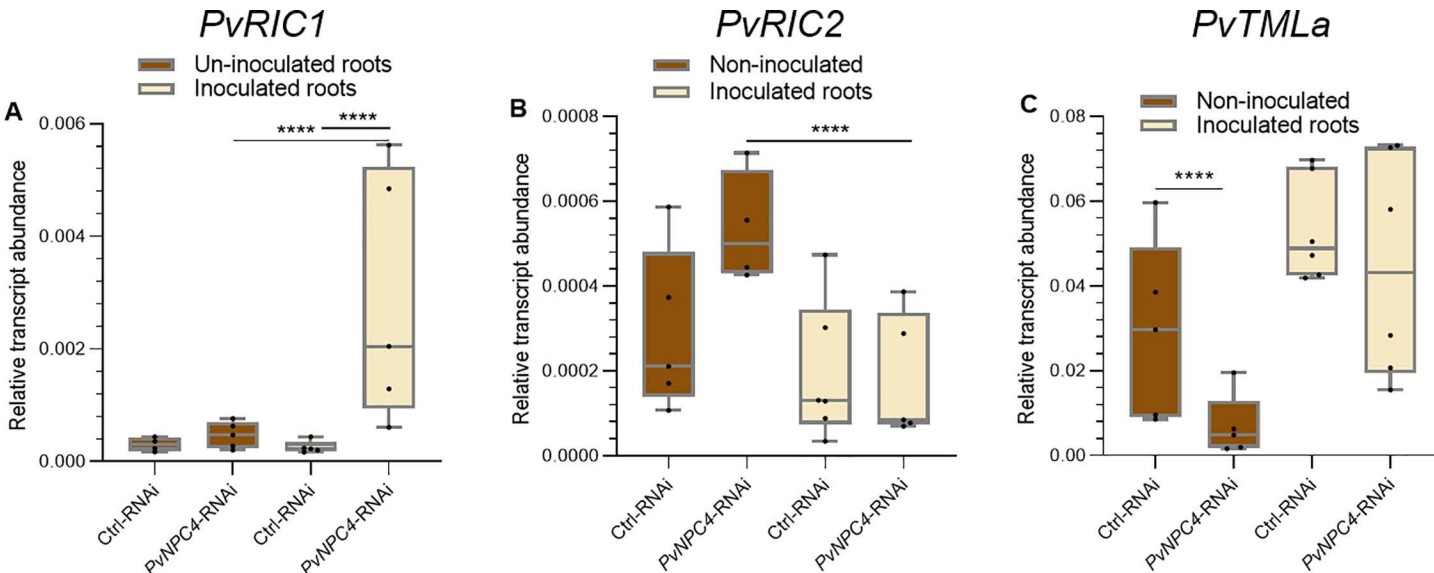

**Fig 8. *PvNPC4* silencing regulates *PvRIC1*, *PvRIC2* and *PvTLMa* transcript abundance in transgenic roots.** Relative transcript abundance of *PvRIC1* **(A)**, *PvRIC2* **(B)**, and *PvTLMa* **(C)**. Ctrl-RNAi indicates transgenic roots carrying the control vector pTdT-SAC. *PvNPC4*-RNAi indicates transgenic roots carrying the *PvNPC4* silencing construct (C1-RNAi and C8-RNAi lines). The lower and upper edges of the boxes delimit the first to third quartiles, respectively, the central horizontal line represents the median, and the whiskers indicate the maximum and minimum values in the data set. For statistical analysis, a Monte Carlo simulation test was used with 9999 samples without replacement (**** $p \leq 0.0001$). Black dots in the box plots indicate individual samples from three biological replicates.

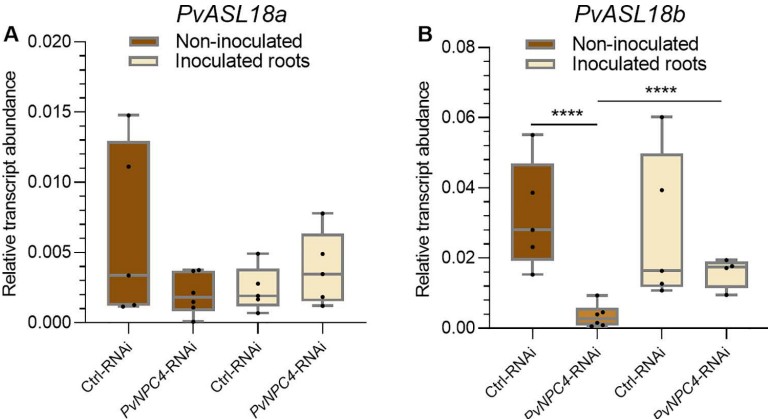

**Fig 9. Silencing of *PvNPC4* does not affect the transcript abundance of *PvASL18a* but does alter that of *PvASL18b*.** Relative transcript abundance of *PvASL18a* **(A)** and *PvASL18b* **(B)**. Ctrl-RNAi indicates transgenic roots carrying the control vector pTdT-SAC. *PvNPC4*-RNAi indicates transgenic roots carrying the *PvNPC4* silencing construct (C1-RNAi and C8-RNAi lines). The lower and upper edges of the boxes delimit the first to third quartiles, respectively, the central horizontal line represents the median, and the whiskers indicate the maximum and minimum values in the data set. For statistical analysis, the Monte Carlo simulation test was used with 9999 samples without replacement (**** $p \leq 0.0001$). Black dots in the box plots indicate individual samples from three biological replicates.

## PLC family in legumes

Our *in silico* analysis of the PLC family in common bean, *L. japonicus*, and barrelclover revealed that legume species generally possess larger numbers of PI-PLC members than NPC members, with the exception of *L. japonicus*, which has equal numbers in the two groups (S2 and S3 Table). This is in contrast to what has been reported in rice and wheat, two cereal species (Poaceae) that have more NPC members [19,20].

Protein sequence analysis revealed that the conserved domain characteristics of PI-PLCs and NPCs in plants [19,20] are preserved in common bean. In particular, all PvPI-PLCs in legumes retain the PLC_X, PLC_Y, and C2 domains. However, the EF-hand motif is not present in all proteins identified in this study, in agreement with the complement of PLCs in other species [19,20,56]. Importantly, the EF-hand motif was present in only four GmPI-PLCs, not in all GmPI-PLCs, as previously reported [22]. Regardless of gene structure, we observed nine exons in *PvPI-PLC* genes and three or four exons in *PvNPC* genes. This finding is in line with the number of exons found in many other species, including chickpea (*Cicer arietinum* L.) [56], soybean [22], rice [19], and wheat [20]. These findings suggest that the gene structure of both *PLC* subfamily genes has been conserved among divergent species.

The phylogenetic tree reconstructed here grouped PI-PLCs and NPCs into two separated clades, regardless of their originating species (Fig 1), consistent with other phylogenetic analyses [18,20,56]. The small phylogenetic distance observed between PvPLCs and GmPLCs (Fig 1) was also supported by a collinearity analysis (Fig 2). Notably, the phylogenetic distance between the PI-PLC clade and the NPC clade was greater than that between the NPC clade and the *M. tuberculosis* PLC used as an out-group. This observation is consistent with what was reported by Nakamura *et al.* (2005) who identified three conserved regions between in *A. thaliana* NPCs and a *M. tuberculosis* PLC.

## Analysis of PvNPC4 function in root development

Analysis of *NPC* transcript abundance in Arabidopsis and rice has revealed that they vary by plant tissue and developmental stage [19,24]. We observed similar patterns for *PvNPC4*, whose transcript abundance fluctuated in roots between developmental stages (S9 Fig) and tissues, being the highest transcript abundance in roots (S10A Fig). This finding suggests a role of *PvNPC4* in nodule development. Indeed, knockdown of *PvNPC4* by RNAi shortened primary roots and resulted in fewer primary and lateral roots (Fig 4). These findings are consistent with the reduction in primary root length observed in two NPC knock-outs of *npc3* and *npc4* from *A. thaliana* [24].

A reduction of root length was also observed in *NPC2* knock-down lines in *A. thaliana* in an *npc6* knock-out background [25]. However, some of the *NPC2* knock-down lines developed an increase in lateral root density, which contrasts with the reduction in lateral roots in *PvNPC4*-RNAi plants. Interestingly, exogenous supplementation of phosphocholine, a product of phosphatidylcholine hydrolysis by NPCs, restores normal root length [25], suggesting that phosphocholine may act as a signal molecule that regulates root development.

On the other hand, our results indicate that the abundance of transcripts of genes involved in auxin signaling was reduced by *PvNPC4* silencing. Particularly, it was observed that the abundance of *PvASL18b* transcripts was reduced in *PvNPC4* roots (Fig 5A). This gene encodes a transcriptional factor, LATERALORGAN BOUNDARIES-DOMAIN 16 (LBD16)/ASYMMETRIC LEAVES2-LIKE 18 (ASL18), that is activated by auxin in the founder cells of lateral roots [3]. In *M. truncatula*, *lbd* mutants showed a reduction in lateral root development of approximately 50%; in contrast, over-expression of an LBD16 gene induced the initiation of ectopic root primordia [55]. In addition, in *L. japonicus*, *asl18a* knock-out developed lower lateral roots density than wild-type plants [57]. Together, these findings suggest that the reduced root development in *PvNPC4*-RNAi composite plants are associated with a regulatory role of *PvASL18b* in auxin signaling.

The involvement of *PvNPC4* in root development in an auxin-dependent manner is also supported by the reduced transcript abundance of the gene encoding the auxin transporter, *PvPin1b*, in *PvNPC4-RNAi* roots (Fig 5B). Phospholipases

$A_2$ are also involved in auxin transport by targeting the PIN protein to the plasma membrane. For instance, knockdown of $PLA_2$ was shown to interfere with trafficking of the PIN protein to the plasma membrane in *A. thaliana* roots. Interestingly, the hydrolytic product of $PLA_2$, lysophosphatidylethanolamine, restored PIN localization to the plasma membrane in a *pla2α* mutant [58]. These results evidence that lipid metabolism is crucial for the accurate auxin-dependent regulatory mechanism. Our findings suggest that *PvNPC4* is a key regulator of root development by regulating auxin signaling.

## Analysis of PvNPC4 function in nodule development

Legumes have developed different mechanisms to strictly control rhizobial symbiosis, with differential gene expression being essential [59–61]. In the present work, *PvNPC4* transcript abundance showed a biphasic pattern being high at early stages of rhizobial symbiosis (24–72 hpi) and low at 14 dpi in inoculated roots and nodules compared to control (Fig 3). This pattern of *PvNPC4* transcript abundance suggests its involvement in rhizobial infection and a putative negative effect on the later mature stages of symbiosis. The decline in *PvNPC4* transcript abundance in inoculated roots and nodules at 14 dpi may reflect a mechanism to prevent phospholipid degradation once nodules have developed. It must be considered that NPCs have non-specific hydrolytic activity towards phospholipids [17]; therefore, their functions may be necessary in the early stages of symbiosis for lipid-mediated signaling that allows infection by rhizobia. At the later and mature stages, when nodules have formed, lipid turnover may be less active; consequently, the hydrolytic activity of PLCs may be more dispensable.

We found a significant decrease in the number of ITs in *PvNPC4*-RNAi roots, which was accompanied by fewer nodules (Fig 6). Previously, it was reported that *PvRbohA* silencing reduced *PvEnod40* transcript abundance and ITs number in common bean [7]. Interesting, our results showed a reduced abundance of *PvEnod40* transcripts in *PvNPC4*-RNAi inoculated roots (Fig 7B). Reduced ITs in common bean was earlier associated with *PvEnod40* transcript abundance [60]; therefore, the drop in *PvEnod40* transcript abundance in *PvNPC4-RNAi* roots may be involved in ITs development.

To further investigate a functional relationship between *PvNPC4* and *PvRbohA*, *PvRbohA* transcript abundance was quantified in *R. tropici*-inoculated and non-inoculated *PvNPC4*-RNAi roots. The slight increase in *PvRbohA* transcript abundance due to *PvNPC4* silencing, regardless of *R. tropici* inoculation (Fig 7C), suggests a role for *PvNPC4* in regulating *PvRbohA* transcription. In this scenario it would be interesting to explore *PvRbohA* transcript abundance at other stages of the symbiosis in a *PvNPC4* background. RBOHs are enzymes that produce superoxide anions, a reactive oxygen species (ROS) [62]; in particular, in common bean, ROS signaling has been shown to respond to rhizobial inoculation and NF perception [63,64]. Furthermore, the *PvRbohA* and *PvRbohB* genes in common bean play key roles during rhizobial symbiosis, as we have described previously [6,7].

ROS homeostasis is very sensitive to rhizobia inoculation and NF perception [63,64] and depending on its levels, can vary according to the stage of development of the symbiosis. Consequently, we suggest that the slight increase in *PvRbohA* transcript abundance observed in *PvNPC4*-RNAi roots may affect proper rhizobial infection. Intriguingly the transcript abundance of *PvNIN,* a master regulator of nodule organogenesis [2] that is also involved in IT formation [65], was not significantly affected by *PvNPC4* silencing at 10 dpi (Fig 7A). However, considering the crucial role of *NIN* in regulating rhizobial symbiosis, its involvement in earlier stages of symbiosis is not discarded. Unfortunately, due to the marked reduction of root development, we had difficulty collecting root tissue at earlier stages.

In the case of the role of *Enod40* in nodule organogenesis, this has been largely documented; for instance, it has been reported to regulate cortical cell division in soybean [66] and the number of nodules in peanut (*Arachis hypogea* L.) [67]. Therefore, the reduction in nodule number could be related to the decrease in *PvEnod40* transcript abundance, in fact, *PvEnod40* transcripts were undetectable in some samples. Considering the increase and decrease of *PvNPC4* transcription in the early and late stages of the symbiosis, respectively, we asked whether the protein product of this gene is involved in the AON pathway. To explore this, transcript abundance of AON-related genes, such as *RIC* and *TML,* was assessed in inoculated and non-inoculate transgenic roots at 10 dpi. Our qPCR analysis shows that *PvNPC* silencing enhanced transcript abundance of *PvRIC1* (Fig 8A). This gene has been shown to be transcriptionally activated by

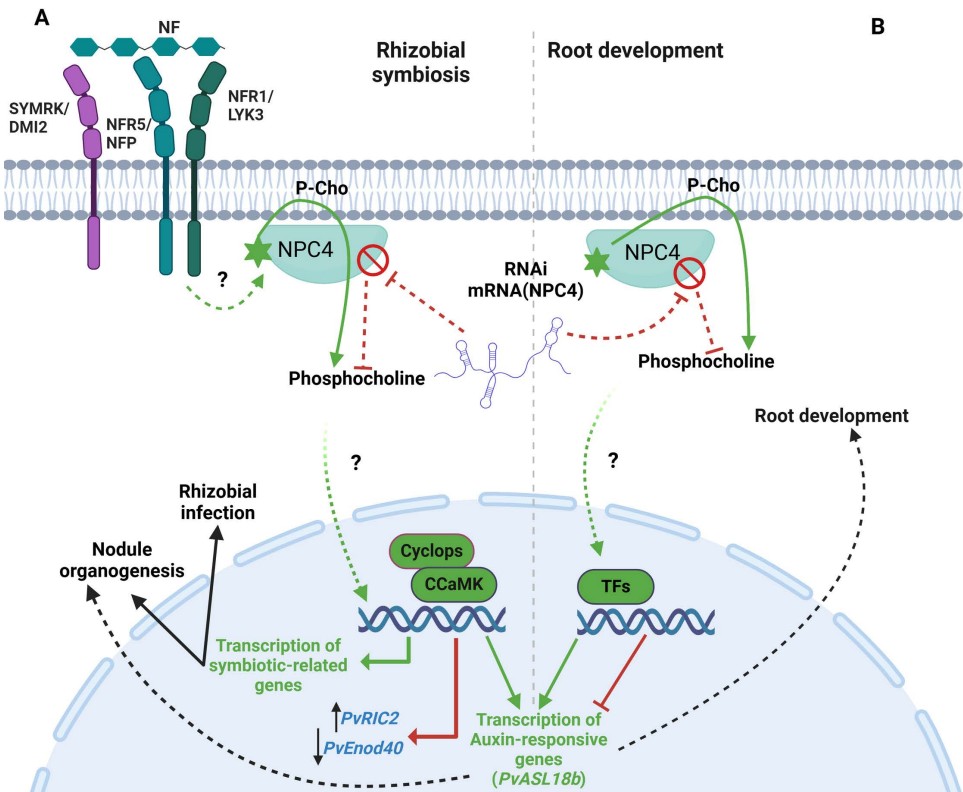

**Fig 10. Proposed model of a signaling cascade in which NPC plays a key role in root and nodule development. (A)** PvNPC is activated, in an unknown manner, in response to NFs secreted by rhizobia (green lines). The hydrolytic activity of PvNPC4 on phosphatidylcholine releases phosphocholine, which is suggested to act as a secondary messenger to induce the transcription of symbiosis-related genes that regulates rhizobial infection and nodule development. At the same time, transcription of auxin response genes is activated, which also regulates rhizobial symbiosis, depending on the concentration and location of auxin. When NPC4 expression is altered by RNAi interference (*PvNPC4*-RNAi in this work, red lines), or by another stimulus, the NPC4 protein level decreases; consequently, the concentration of phosphocholine decreases. Therefore, the positive role of NPC4 in regulating nodule development fails or is attenuated. Our results showed that under these conditions, the transcript abundance of *PvRIC1* and *PvEnod40* is modulated, which may regulate nodule development. **(B)** Under normal conditions, NPC plays a positive regulatory role in root development (green lines), probably mediated by phosphocholine. If NPC expression is decreased, as explained above (red lines), transcription of genes involved in auxin signaling, a master regulator of root development is inhibited. This would provoke a decrease in root development. Our results indicated that *PvASL18b* is involved in this mechanism. In summary, our results suggest that *PvNPC4* modulates both root and nodule development by regulating auxin-mediated signaling (black dashed arrows).

inoculation of *R. phaseoli* into common white bean [10], another variety of *P. vulgaris*, indicating an important role of the RIC1 protein in the AON mechanism in *P. vulgaris*.

*RIC* genes encode CLE peptides in response to rhizobia, initiating the AON. These peptides are transported through the xylem to the shoot where they are recognized by leucine-rich repeats receptor-like kinases (LRR-RLKs) that inhibit the production of miR2111. [54]. In *M. truncatula* miR2111 acts as a shoot-to-root signaling that negatively regulates nodule *TML* transcript abundance in roots, thereby negatively regulating nodule development [68]. In this scenario, the induction of *PvRIC1* in inoculated *PvNPC4*-RNAi roots is consistent with the reduction of nodule number caused by *PvNPC4* silencing (Fig 6B). Our results, together with the reported findings on the AON mechanism, suggest that PvNPC4 acts as a positive regulator of nodulation upstream *RIC* transcription.

The low transcript abundance of *PvRIC2* in inoculated transgenic roots suggests that this gene is not induced immediately after *R. tropici* inoculation (10 dpi). However, it is intriguing that *PvRIC2* transcript abundance increased in

non-inoculated *PvNPC4*-RNAi roots and decreased in those inoculated with this difference being significant (Fig 8B). These observations point out that the *PvNPC4* protein regulates *PvRIC2* but, probably, at 10 dpi, PvRIC2 was not required for AON signaling, as was PvRIC1. Therefore, *PvRIC2* reduction could be a compensatory mechanism of *PvRIC* transcription for the precise signaling function of AON.

On the other hand, a reduction of *PvTMLa* transcript abundance was observed in non-inoculated *PvNPC4*-RNAi roots, suggesting a role for this gene in the AON mechanism in common bean. Interestingly, this effect of PvNPC4 on *PvTMLa* transcription was not observed in inoculated roots (Fig. 8C). Interestingly, *R. tropici* inoculation induced a slight increase in *PvTMLa* transcript abundance in *PvNPC4*-RNAi and RNAi control inoculated roots compared to their non-inoculated counterparts, indicating the involvement of PvTMLa in the AON mechanism in common bean. However, the role of PvTMLa in the common bean-*R. tropici* symbiosis appears not to be regulated by PvNPC4 at 10 dpi. The common bean genome encodes five *TML* genes, where *PvTMLa* has the highest transcript abundance in inoculated roots and in nodules at 5 and 15 dpi (S13 Fig). This prompted us to explore *PvTMLa* transcription in inoculated and non-inoculated transgenic roots. The participation of the rest of the *PvTML* genes in the regulation of this symbiosis by PvNPC4 is of interest for future research. To our knowledge, this is the first exploration of the function of *TML* genes in common bean.

The *LOB16/ASL18* genes have been shown to be involved in both root and nodule development [52,55]. Our results show that *PvNPC4* silencing did not affect *PvASL18a* transcript abundance regardless of *R. tropici* inoculation (Fig 9A), suggesting a non-functional relationship of PvNPC4 and PvASL18a in root and nodule development. As observed in Fig 9B, PvASL18b seems to be involved in the role of PvNPC4 in root development, but not in nodule development.

## Conclusions and perspectives

In the present study, we provide evidence that downregulation of *PvNPC4* transcript abundance negatively affected root development in this legume species. Interestingly, this effect appears to be related to auxin transport, as the transcript abundance of genes involved in auxin signaling decreased significantly or slightly in *PvNPC4*-RNAi roots, depending on the gene analyzed. More detailed studies using auxin markers to elucidate dynamic changes in auxin transport and synthesis would be valuable for a better understanding of PvNPC4-mediated regulation of auxin-dependent root development. Further studies may be needed to examine the potential seedling lethality of *PvNPC4* knockouts, as well as functional redundancy among *PvNPC* members.

The establishment of rhizobial symbiosis involves several biological processes, such as membrane turnover, lipid-mediated signaling and metabolism, and autoregulation of nodulation (AON). NPCs are enzymes closely involved in membrane remodeling and lipid metabolism due to their hydrolytic activity towards phospholipids. Furthermore, lipid signaling is a key signaling mechanism for the regulation of various functions in plants. We showed that downregulation of *PvNPC4* decreased rhizobial infection in common bean hairy roots with a subsequent drop in nodule number. Importantly, our results show evidence for the involvement of PvNPC4 in the AON. To the best of our knowledge, this is the first report on the involvement of a PLC in rhizobial symbiosis.

In summary, we showed here that silencing *PvNPC4* results in shorter and fewer roots, as well as fewer nodules. It is important to emphasize that a conserved regulatory pathway shared between lateral roots and nodule development has been suggested and discussed previously [3,57]. Therefore, we hypothesize that PvNPC4 might be involved in this shared pathway involving auxin signaling, as shown in the model illustrated in Fig 10.

## Supporting information

**S1 File. Supplementary tables.**
(DOCX)

**S2 File. Supplementary figures.**
(DOCX)

**S3 File. In silico data.**
(XLSX)

**S4 File. Raw data.**
(XLSX)

**S5 File. Heatmaps.**
(PDF)

## Acknowledgments

We thank Ph. D. Citlali Fonseca-García for critical reading of the manuscript. We also thank B. Sc Mariel Escobar, B. Sc. Marlén Delgado, and B. Sc. Mary Jose Enriquez for their support in the experiments carried out in the greenhouse and Ph. D. Javier Montalvo-Arredondo for the support in image preparation. In addition, we thank Unidad de Síntesis y Secuenciación at Instituto de Biotecnología, UNAM for technical support regarding oligonucleotide synthesis and DNA sequencing.

## Author contributions

**Conceptualization:** Ronal Pacheco, Carmen Quinto.

**Data curation:** Ronal Pacheco, MA Juárez-Verdayes, A. I. Chávez-Martínez, Janet Palacios-Martínez, Alfonso Leija, Noreide Nava.

**Formal analysis:** Ronal Pacheco, MA Juárez-Verdayes, Carmen Quinto.

**Funding acquisition:** Carmen Quinto.

**Investigation:** Ronal Pacheco, MA Juárez-Verdayes.

**Methodology:** Ronal Pacheco, MA Juárez-Verdayes, A. I. Chávez-Martínez, Janet Palacios-Martínez, Alfonso Leija.

**Project administration:** Carmen Quinto.

**Resources:** Carmen Quinto.

**Supervision:** Carmen Quinto.

**Validation:** Carmen Quinto.

**Visualization:** Ronal Pacheco, Carmen Quinto.

**Writing – original draft:** Ronal Pacheco, MA Juárez-Verdayes.

**Writing – review & editing:** Ronal Pacheco, Luis Cárdenas-Torres, Carmen Quinto.

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
