## [Decision Letter · Decision Letter 0]

20 Oct 2024

PONE-D-24-24822Genomic and functional characterization of the phospholipase C family during rhizobia–common bean symbiosisPLOS ONE

Dear Dr. Quinto,

Thank you for submitting your manuscript to PLOS ONE. After careful consideration, we feel that it has merit but does not fully meet PLOS ONE’s publication criteria as it currently stands. Therefore, we invite you to submit a revised version of the manuscript that addresses the points raised during the review process.

We look forward to receiving your revised manuscript.

Kind regards,

Ying Ma, Ph.D.

Academic Editor

PLOS ONE

2. Thank you for stating the following financial disclosure: [RP: 749422, Consejo Nacional de Ciencia y Tecnología. https://conahcyt.mx/ IN204024, Programa de Apoyo a Proyectos de Investigación e Innovación Tecnológica/UNAM.  https://dgapa.unam.mx/index.php/impulso-a-la-investigacion/papiit CQ: CF-2023-I-297, Consejo Nacional de Ciencia y Tecnología in México.  https://conahcyt.mx/ IN203021,  Programa de Apoyo a Proyectos de Investigación e Innovación Tecnológica/UNAM. https://dgapa.unam.mx/index.php/impulso-a-la-investigacion/papiit]. Please state what role the funders took in the study. If the funders had no role, please state: "The funders had no role in study design, data collection and analysis, decision to publish, or preparation of the manuscript." If this statement is not correct you must amend it as needed. Please include this amended Role of Funder statement in your cover letter; we will change the online submission form on your behalf.

3. Please include a caption for figure Fig 4A, Fig 5, Fig 7A.

Additional Editor Comments (if provided):

Reviewers' comments:

Reviewer's Responses to Questions

**Comments to the Author**

1. Is the manuscript technically sound, and do the data support the conclusions?

Reviewer #1: Yes

Reviewer #2: Yes

2. Has the statistical analysis been performed appropriately and rigorously? 

Reviewer #1: Yes

Reviewer #2: Yes

3. Have the authors made all data underlying the findings in their manuscript fully available?

Reviewer #1: Yes

Reviewer #2: Yes

4. Is the manuscript presented in an intelligible fashion and written in standard English?

Reviewer #1: Yes

Reviewer #2: Yes

5. Review Comments to the Author

Reviewer #1: In this manuscript, the authors reported data supporting the symbiotic role of PvNPC4 in P. vulgaris. They demonstrated that the downregulation of PvNPC4 affects root development and nodule formation. The presented data advance our knowledge about the role of NPC in the root nodule symbiosis.

Comments:

Could you please elaborate on the criteria used to select PvPI-PLC4 and PvNPC4? This information is crucial for a comprehensive understanding of your research.

Do the authors observe any defect in root hairs? Could defects in root development, particularly lateral roots, compromise nodule development?

Is it possible that negative regulators of the nodule formation (i.e., AON pathway) are active in PvNPC4-RNAi roots? It would be interesting to assess the expression of RIC and TML in PvNPC4-RNAi roots.

The authors must remember that different genetic modules regulate the rhizobial infection and nodule development programs. Indeed, some mutants form several infection threads but no nodules. Please modify this in the Discussion section.

I strongly recommend that you include a clear depiction of the root phenotype as a main figure. This will greatly enhance the visual understanding of your research.

Reviewer #2: Data on manuscript are strongly supported by a careful experimental procedure and strong statistics approaches. The data obtained supports clearly and strongly the obtained results. The in silico analysis of the PLC genes, gives a genomic supports of ancient divergence between leguminous and non leguminous plants and close relationship between bean and soybean species.

The paper shows evidence of the number and identity of the PLC genes family in bean and confirms the transcription pattern of root and nodule of PLC related sequences, allowing to select PLC genes involved in root and symbiotic development and function.

The results obtained when PvNPC4 is silenced by RNA interference, demonstrated the participation of this gene in the development of primary and lateral roots. Also the knockdown of PvNPC4 reduce the infection process and the nodule development during the symbiotic interaction of bean with rhizobia.

The functional analysis by PvNPC4-RNAi shows by the first-time evidence of its involvement in bean root development and the nitrogen fixing symbiosis.

This results open a new area in order to further understand the molecular mechanism implicated in the symbiotic process through lipid signaling.

Minor ortographic or typographic details:

Materials and methods:

Lotus japonicus), (Arabidopsis, (Glycine max)

transcript abundance of PvPC4 (PvPNPC4)

transcriptional profile of PvPC4 (PvPNPC4)

Fig 2. …. gene duplation (duplication) …

S10 Fig Figure title PvPC4 (PvNPC4) (Phvul.010G033400.1)

6. PLOS authors have the option to publish the peer review history of their article (what does this mean? ). If published, this will include your full peer review and any attached files.

**Do you want your identity to be public for this peer review?** For information about this choice, including consent withdrawal, please see our Privacy Policy .

Reviewer #1: No

Reviewer #2: No

---

## [Author Response · Author response to Decision Letter 0]

4 Dec 2024

Review Comments to the Author

Reviewer #1: In this manuscript, the authors reported data supporting the symbiotic role of PvNPC4 in P. vulgaris. They demonstrated that the downregulation of PvNPC4 affects root development and nodule formation. The presented data advance our knowledge about the role of NPC in the root nodule symbiosis.

Comments:

-Could you please elaborate on the criteria used to select PvPI-PLC4 and PvNPC4? This information is crucial for a comprehensive understanding of your research.

R. We appreciate the comment. Our criteria for selecting PvPI-PLC4 was initially based on its transcription profile in PvGEA (Common Bean Gene Expression Atlas (https://www.zhaolab.org/PvGEA/) (O’Rourke et al., 2014). As observed in Fig. S7. PvPI-PLC4 has the highest level of expression (RPKM) in roots and nodules. The PvGEA data are very useful as a reference for our studies; however, we always do qPCR of the gen of interest to corroborate the abundance of the transcripts of interest. Therefore, we performed qPCR on our samples and found barely detectable levels of the PvPI-PLC4 transcript. For this reason, we discarded PvPI-PLC4 for further analyses. On the other hand, PvNPC4 is the ortholog of A. thaliana NPC4, which was previously characterized and reported that is a plasma membrane-localized protein and has a key role in degradating plasma membrane phospholipids (Nakamura et al., 2005). Since phospholipid metabolism and membrane lipid turnover are essential for nodule development, we were encouraged to study the function of PvNPC4 in common bean-R. tropici symbiosis. These criteria are described in lines 325-332.

-Do the authors observe any defect in root hairs?

R. Unfortunately we were not able to analyze the root hairs microscopically due to several technical difficulties when performing that experiment. We have discussed this issue in the discussion section (lines: 640-641).

-Could defects in root development, particularly lateral roots, compromise nodule development?

-R. We appreciate the question. We consider that the mechanisms of root and nodule formation are correlated. In fact, previous studies have demonstrated the participation of some genes in the regulation of the development of both roots and nodules (Schiessl et al., 2019; Soyano et al., 2019). It is proposed that during evolution, some of the components of root developmental pathways have been recruited in nodule development (Soyano et al., 2021). In this version of the manuscript, we analyze and discuss the effect of PvNPC4 silencing on the transcript abundance of the two LATERAL ORGAN BOUNDARIES-DOMAIN 16 (LBD16)/ASYMMETRIC LEAVES2-LIKE 18 (ASL18) genes involved in root development. The transcript abundance of one these genes (PvASL18b) was significantly reduced in PvNPC4-RNAi roots (see lines: 417-418 and Fig 5A). Likewise, the transcript abundance of a gene encoding an auxin transporter (PvPin1b) was also reduced by PvNPC4 silencing (see lines: 419-421 and Fig 5B). Finally, we propose a model that explains the role of PvNPC4 in root and nodule development (Fig 10).

-Is it possible that negative regulators of the nodule formation (i.e., AON pathway) are active in PvNPC4-RNAi roots? It would be interesting to assess the expression of RIC and TML in PvNPC4-RNAi roots.

R. Thanks for the question. In this new version of the manuscript, we analyzed the effect of PvNPC4 silencing on the transcript abundance of several genes involved in AON, such as PvRIC1, PvRIC2, and PvTMLa. The results obtained showed an increase in the abundance of the PvRIC1 transcript, suggesting the activation of the AON pathway. See the results obtained in lines 487-496 and Fig 8A, B.

-The authors must remember that different genetic modules regulate the rhizobial infection and nodule development programs. Indeed, some mutants form several infection threads but no nodules. Please modify this in the Discussion section.

R. We agree with the observation. The discussion section has been modified as indicated throughout that section (lines 614-620 and 642-646).

-I strongly recommend that you include a clear depiction of the root phenotype as a main figure. This will greatly enhance the visual understanding of your research.

R. Thanks for the recommendation. A figure illustrating the root phenotype was included as the main figure in this version of the manuscript (Fig. 4).

Reviewer #2: Data on manuscript are strongly supported by a careful experimental procedure and strong statistics approaches. The data obtained supports clearly and strongly the obtained results. The in silico analysis of the PLC genes, gives a genomic supports of ancient divergence between leguminous and non leguminous plants and close relationship between bean and soybean species.

R. We really appreciate the comment. We consider that proper statistical analysis is crucial to improve data interpretation and visualization.

The paper shows evidence of the number and identity of the PLC genes family in bean and confirms the transcription pattern of root and nodule of PLC related sequences, allowing to select PLC genes involved in root and symbiotic development and function.

R. The transcriptomic data available in the repository are useful for selecting candidate genes to study in a specific process of interest, even more so when the gene family has not been reported.

The results obtained when PvNPC4 is silenced by RNA interference, demonstrated the participation of this gene in the development of primary and lateral roots. Also the knockdown of PvNPC4 reduce the infection process and the nodule development during the symbiotic interaction of bean with rhizobia.

R. We appreciate the comment. Our data reveal the role of PvNPC4 in root and nodule development, providing further evidence for shared pathways to regulate root and nodule development, as reported (Schiessl et al., 2019; Soyano et al., 2019) and discussed earlier (Lebedeva et al., 2021; Soyano et al., 2021)

The functional analysis by PvNPC4-RNAi shows by the first-time evidence of its involvement in bean root development and the nitrogen fixing symbiosis.

R. Thank for the involvement of PLC in rhizobial symbiosis. We believe that this report can contribute to the understanding of the molecular mechanisms underlying the establishment of this symbiosis.

This results open a new area in order to further understand the molecular mechanism implicated in the symbiotic process through lipid signaling.

R. We really appreciate this comment. We are excited to further analyze and study the role of lipid-mediate signaling during rhizobial symbiosis.

Minor ortographic or typographic details:

Materials and methods:

Lotus japonicus), (Arabidopsis, (Glycine max)

transcript abundance of PvPC4 (PvPNPC4)

transcriptional profile of PvPC4 (PvPNPC4)

Fig 2. …. gene duplation (duplication) …

S10 Fig Figure title PvPC4 (PvNPC4) (Phvul.010G033400.1)

R. Thanks for the correction. These typographical errors were amended in the new version.

Geneal comments

Throughout the text, we have changed expressions such as: expression levels, transcript levels, and expression, for “transcript abundance”, since this is a more precise expression to refer to the results of qPCR analyses.

---

## [Decision Letter · Decision Letter 1]

22 Dec 2024

The non-specific phospholipase C of common bean PvNPC4 modulates roots and nodule development

PONE-D-24-24822R1

Dear Dr. Quinto,

We’re pleased to inform you that your manuscript has been judged scientifically suitable for publication and will be formally accepted for publication once it meets all outstanding technical requirements.

Kind regards,

Ying Ma, Ph.D.

Academic Editor

PLOS ONE

Additional Editor Comments (optional):

Reviewers' comments:

Reviewer's Responses to Questions

**Comments to the Author**

1. If the authors have adequately addressed your comments raised in a previous round of review and you feel that this manuscript is now acceptable for publication, you may indicate that here to bypass the “Comments to the Author” section, enter your conflict of interest statement in the “Confidential to Editor” section, and submit your "Accept" recommendation.

Reviewer #1: All comments have been addressed

2. Is the manuscript technically sound, and do the data support the conclusions?

Reviewer #1: Yes

3. Has the statistical analysis been performed appropriately and rigorously? 

Reviewer #1: Yes

4. Have the authors made all data underlying the findings in their manuscript fully available?

Reviewer #1: Yes

5. Is the manuscript presented in an intelligible fashion and written in standard English?

Reviewer #1: Yes

6. Review Comments to the Author

Reviewer #1: All my comments were addressed. The revised version reads better and will contribute to the field of the root nodule symbiosis

7. PLOS authors have the option to publish the peer review history of their article (what does this mean? ). If published, this will include your full peer review and any attached files.

**Do you want your identity to be public for this peer review?** For information about this choice, including consent withdrawal, please see our Privacy Policy .

Reviewer #1: No

---

## [Editor Report · Acceptance letter]

PONE-D-24-24822R1

PLOS ONE

Dear Dr. Quinto,

I'm pleased to inform you that your manuscript has been deemed suitable for publication in PLOS ONE. Congratulations! Your manuscript is now being handed over to our production team.

Kind regards,

on behalf of

Dr. Ying Ma

Academic Editor

PLOS ONE